# The impact of HTLV-1 expression on the 3D structure and expression of host chromatin

**Hiroko Yaguchi, Anat Melamed, Saumya Ramanayake, Helen Kiik, Aviva Witkover, Charles R. M. Bangham** \*

Department of Infectious Diseases, Faculty of Medicine, Imperial College London, London, United Kingdom

\* c.bangham@imperial.ac.uk

## Abstract

A typical HTLV-1-infected individual carries >$10^4$ different HTLV-1-infected T cell clones, each with a single-copy provirus integrated in a unique genomic site. We previously showed that the HTLV-1 provirus causes aberrant transcription in the flanking host genome and, by binding the chromatin architectural protein CTCF, forms abnormal chromatin loops with the host genome. However, it remained unknown whether these effects were exerted simply by the presence of the provirus or were induced by its transcription. To answer this question, we sorted HTLV-1-infected T-cell clones into cells positive or negative for proviral plus-strand expression, and then quantified host and provirus transcription using RNA-seq, and chromatin looping using quantitative chromosome conformation capture (q4C), in each cell population. We found that proviral plus-strand transcription induces aberrant transcription and splicing in the flanking genome but suppresses aberrant chromatin loop formation with the nearby host chromatin. Reducing provirus-induced host transcription with an inhibitor of transcriptional elongation allows recovery of chromatin loops in the plus-strand-expressing population. We conclude that aberrant host transcription induced by proviral expression causes temporary, reversible disruption of chromatin looping in the vicinity of the provirus.

**Data Availability Statement:** Sequence data have been deposited at the European Nucleotide Archive, (ENA, https://www.ebi.ac.uk/ena/browser/home), accession number PRJEB71982.

## Author summary

The human T cell leukemia virus HTLV-1 causes an aggressive leukemia or lymphoma in ~5% of people infected with the virus, and a further 1–4% develop a chronic inflammatory disease that leads to progressive paralysis of the legs. HTLV-1 is a retrovirus, like HIV, and these viruses insert themselves–as a 'provirus'—into the DNA of the T lymphocytes that they infect: this is largely why the viruses are very difficult to eradicate. The provirus remains dormant for most of the time, but it can be reactivated when the T cell meets a change in temperature or pH, or a number of other conditions.

We recently discovered that HTLV-1, when it is integrated into the host DNA, changes the 3D structure of the genome in the infected cell, and interferes with the normal function of the host genes that lie near HTLV-1 in the genome. What we have now found is that the change in the 3D structure of the genome is caused simply by the presence of the

**Funding:** This work was supported by a Wellcome Trust UK Investigator Award to C.R.M.B. (ref. 207477). The funders had no role in study design, data collection and analysis, decision to publish, or preparation of the manuscript.

**Competing interests:** The authors have declared that no competing interests exist.

provirus, but that when the provirus is reactivated the 3D structural changes are temporarily removed, and nearby host genes can be abnormally activated by the provirus.

## Introduction

The 3D structure of chromatin is important in many biological processes, including transcription regulation, DNA replication, cell cycle and differentiation. The CCCTC-binding factor (CTCF) is a key player in chromatin loop formation [1,2]. CTCF binds a non-palindromic 20-nucleotide DNA motif at ~50,000 sites in the human genome [2]; two CTCF molecules bound to different genomic sites can dimerize, forming the boundary of chromatin loops extruded through the cohesin complex. Chromatin looping plays a critical part in gene regulation by regulating the contacts between specific enhancers and promoters.

Human T cell leukemia virus type 1 (HTLV-1, also known as human T-lymphotropic virus type 1), mainly infects CD4[+] T-cells *in vivo*. Most carriers remain asymptomatic, but in 5% to 10% of the infected individuals, HTLV-1 infection leads to either an aggressive T-cell malignancy known as adult T-cell leukemia/lymphoma (ATL), or a chronic progressive neuro-inflammatory condition called HTLV-1-associated myelopathy/tropical spastic paraparesis (HAM/TSP; referred to hereafter as HAM) [3]. HTLV-1 persists in the host mainly by clonal proliferation of infected cells, and a typical host carries $>10^4$ long-lived HTLV-1-infected T-cell clones [4]: each clone can be distinguished by the unique integration site of the single-copy provirus in the host genome [5].

The HTLV-1 provirus has two strands (**Fig 1**). The plus strand, transcribed from the 5′ LTR, encodes the structural proteins, the transcriptional transactivator protein Tax, the regulator of mRNA splicing and transport Rex, and minor accessory proteins. The minus strand, transcribed from the 3′ LTR, encodes the regulatory protein HBZ [6].

Single-cell heterogeneity in HTLV-1 proviral expression has been shown in naturally HTLV-1-infected T cell clones [7–9]. At a given time, a small proportion of cells express intense bursts of *tax*, the proportion varying between the clones [7] **(Tables 1 and 2).** By contrast, the minus strand is expressed in approximately 50% of circulating infected T cells at a given time [8]. Each HTLV-1-infected T-cell clone has its own pattern of proviral expression, that is, the frequency, intensity and duration of the transient transcriptional burst of the proviral plus-strand [10–12]. Since the HTLV-1 proviral sequence varies little within the host, these clone-specific differences are thought to be largely due to the unique genomic integration site of the provirus; other factors may include the antigen specificity and epigenetic modifications of the host cell.

Disorganization of 3D chromatin structure can cause diseases by rewiring interactions between genes and regulatory elements [13], and certain mutations in CTCF are linked with human disease [14–16]. We previously showed that the HTLV-1 provirus contains a CTCF binding site (BS): the provirus creates novel loops with the host genome, inducing transcriptional deregulation in the host genome flanking the provirus [10,17]. However, it remained unknown whether these effects were exerted simply by the presence of the provirus or by its transcription.

To answer this question, we sorted naturally-infected HTLV-1[+] T cell clones to obtain subsets of provirus-expressing cells and non-expressing cells, and analysed the effects of proviral expression on chromatin looping, using quantitative circular chromosome conformation capture (q4C) [10], and on host transcription, using RNA-seq.

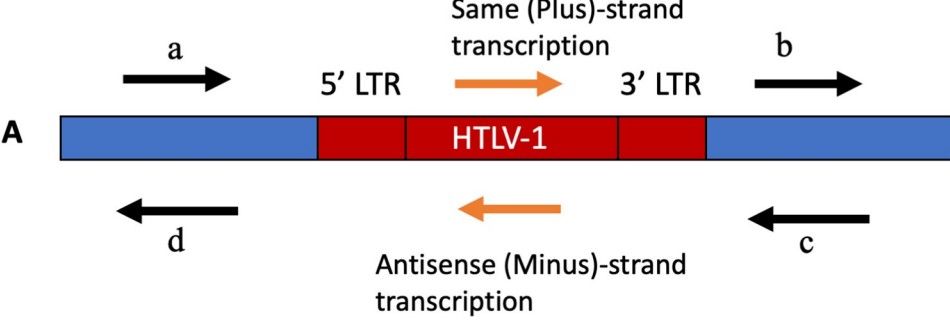

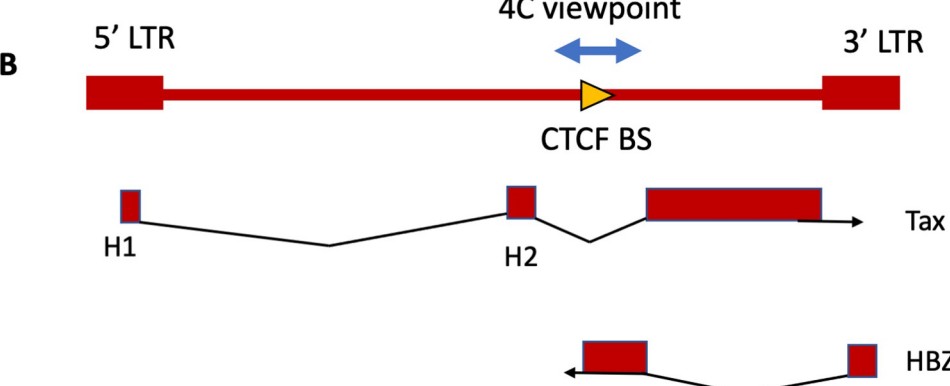

**Fig 1. Structure and expression of the HTLV-1 provirus in the host genome.** (A) Definitions regarding classification of the direction of transcription. Here we use 'same sense' to denote transcription from the same strand of the host genome as the proviral plus strand (e.g. *tax* gene). We refer to flanking host transcription upstream and downstream of 5′LTR-driven plus-strand proviral expression respectively as (a) same (plus) sense, 5′ side of the provirus and (b) same sense, 3′ side of the provirus. Similarly, we refer to transcription upstream and downstream of 3′LTR-driven minus-strand proviral expression respectively as (c) antisense, 3′ side of the provirus and (d) antisense, 5′ side of the provirus. In this study we focus on the effects of plus-strand expression. (B) Diagram of HTLV-1 proviral genome and splicing pattern of the regulatory genes *tax* (encoded in plus strand) and *HBZ* (encoded in minus strand); red boxes represent exons. The 4C viewpoint containing the CTCF binding site (BS) is shown.

## Results

### HTLV-1 proviral plus-strand expression reduces chromatin looping between the provirus and the host genome

We hypothesized that the chromatin loops between the provirus and host genome region are regulated by expression of the provirus. To test this hypothesis, we focused on the proviral plus-strand, which is expressed in intense intermittent transcriptional bursts [7]. We used the HTLV-1 transcriptional transactivator protein Tax as a marker of plus-strand expression. T cell clones naturally infected with HTLV-1, each of which has a unique integration site (**Table 1**), were crosslinked with paraformaldehyde, flow-sorted into Tax$^+$ and Tax$^-$ populations (**S1 Fig**), and subjected to q4C assay, using a fragment of the proviral sequence containing its CTCF binding site as the q4C viewpoint [10] (Fig 1B).

**Table 1. Clone list.**

| Clone | Genomic location of dominant integration site, hg 19* | Provirus orientation ** | Subject | Clone derived from | tax expression reporter | Reference |
|---|---|---|---|---|---|---|
| HA1 | chr07:18905519 | R | HEZ | Asymptomatic carrier | - | Newly established |
| 11.63 | chr19:33829548 | F | TBW | HAM/TSP patient | - | [5,10] |
| TBX4B | chr22:44323198 | F | TBX | HAM/TSP patient | - | [5,10] |
| 3.60 | chr04:70567285 | F | TBJ | HAM/TSP patient | - | [5,10] |
| d2EGFP-TBX4B | chr22:44323198 | F | TBX | HAM/TSP patient | GFP | [12] |
| d2EGFP-11.63 | chr19:33829548 | F | TBW | HAM/TSP patient | GFP | Newly established |
| d2EGFP-11.50 | chr19:28282587 | R | TBW | HAM/TSP patient | GFP | [11] |
| Timer -TBX4B | chr22:44323198 | F | TBX | HAM/TSP patient | Timer protein | [11] |
| Timer- 3.60 | chr04:70567285 | F | TBJ | HAM/TSP patient | Timer protein | [11] |

* hg 19 denotes human Genome Build 19

** Orientation of the provirus relative to the hg19 chromosome coordinates. F: provirus is oriented 5′ to 3′ in the chromosomal plus-strand. R: provirus is oriented 5′ to 3′ in the chromosomal minus-strand.

In the Tax⁻ population of clones 3.60 and HA1, we identified q4C peaks (long-range chromatin loops between the provirus and the host genome) (**Fig 2A**)**:** the identified peaks often overlapped CTCF binding sites, as previously reported in the study using unsorted T cells [10].

Unexpectedly, the frequency of the chromatin loops identified in Tax⁻ cells was much lower in Tax⁺ cells, although the technical peak at the viewpoint (VP), which is always present in a successful 4C-seq assay [18], remained in Tax⁺ cells (**Fig 2A**).

Tax protein is a transcriptional transactivator that stimulates transcription of both HTLV-1 proviral genes and many host genes. To exclude the possibility that the anti-Tax antibody used in intracellular staining altered chromatin looping, we performed q4C assays using T-cell clones transduced with a reporter construct which expresses a modified EGFP with a half-life of ~2h (d2EGFP-TBX4B) when stimulated by Tax protein [11]. The GFP signal intensity was positively correlated with Tax expression [12]. Provirus-expressing (GFP⁺) cells and non-expressing (GFP⁻) cells were flow-sorted after fixation in 1% paraformaldehyde and subjected to the q4C assay. Consistent with the results from intracellular Tax staining (Fig 2A), GFP⁺ cells also showed fewer chromatin loops compared with GFP⁻ cells (**S2 Fig**), and the 4C

**Table 2. Tax expression of HTLV-1-infected T cell clones.**

| Clone | Tax-positive (%) |
|---|---|
| 3.60 | 4 ± 0.8 |
| TBX4B | 8 ± 0.1 |
| 11.63 | 36 ± 5.5 |
| 11.50 | 40 ± 2.5 |
| HA1 | 56 ± 3.7 |

T cell clones (clone details as shown in Table 1) were stained for Tax protein and analysed by flow cytometry. Mean ± SD (n = 2 or 3 biological replicate experiments) of Tax-positive fraction of viable cells are shown. Note: The reporter clones (GFP or Timer) shown in Table 1 were established from clones shown in this table. While the transcriptional burst of tax is short-lived, Tax protein persists for some days in each cell. In the present experiment, the cells were selected on the basis of Tax protein expression: at any one time, therefore, the percentage of Tax protein-positive cells at any instant in these clones in vitro is relatively high [11,12].

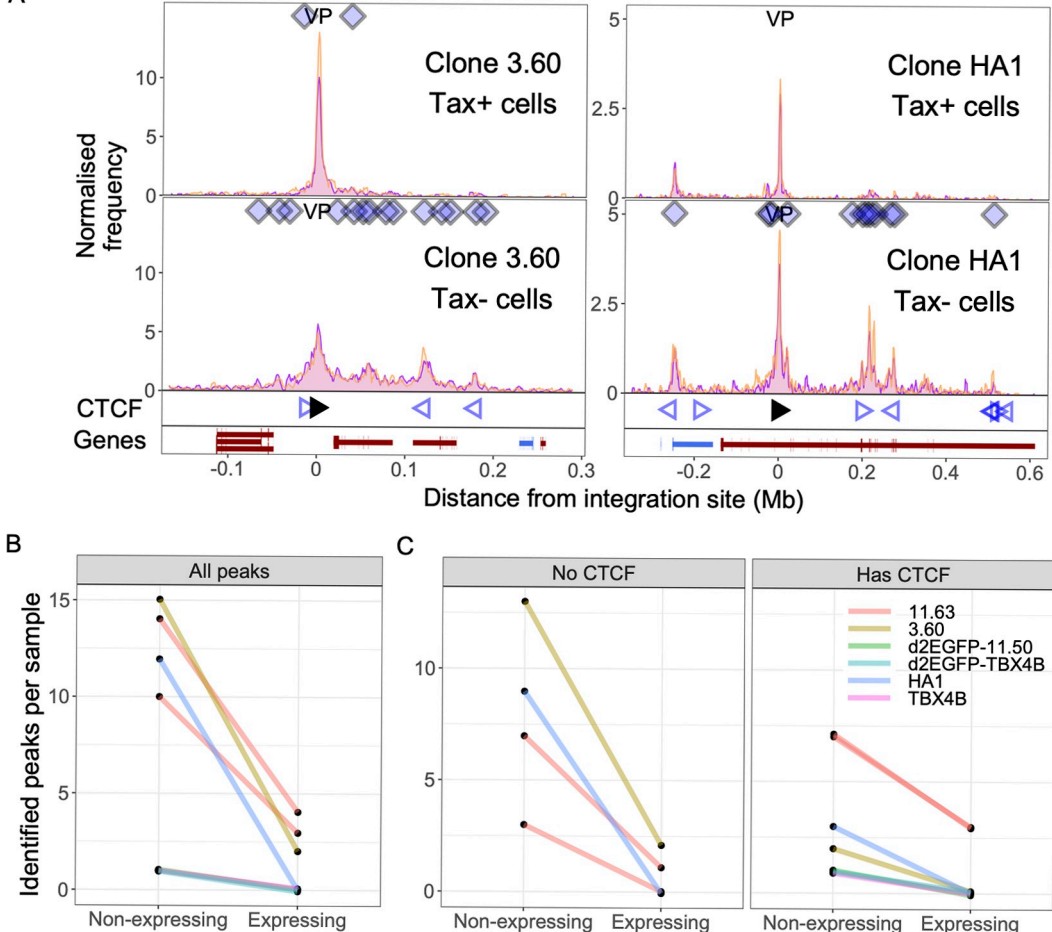

**Fig 2. HTLV-1 plus-strand expression results in fewer chromatin loops (number of q4C peaks) between provirus and host genome.** (A) q4C profiles of Tax⁻and Tax⁺ cells of two different clones (3.60 and HA1). For each clone, the top panel depicts the q4C profile in the 5′ and 3′ host genome flanking the provirus (two biological duplicates), quantified as the normalized frequency of ligation events in overlapping windows (window width 10 kb, step 1 kb). On the horizontal axis, positive values denote positions extending from the 3′ LTR side of the provirus; negative values denote positions 5′ of the 5′LTR. VP–viewpoint in q4C (proviral integration site). Diamonds mark the positions of reproducible chromatin contact sites identified by the peak calling algorithm. CTCF track–open arrowheads denote positions of CTCF-binding sites (BS); the filled arrowhead denotes the CTCF-BS in the provirus. Genes track shows RefSeq protein-coding genes in the flanking host genome. The q4C profiles of remaining clones are shown in S2 and S3 Figs. (B) Number of identified peaks in non-expressing (Tax- or GFP-) and expressing (Tax⁺ or GFP⁺) subsets isolated from 6 clones: total (all peaks) and (C) peaks with or without a CTCF binding site.

profiles of GFP⁺ d2EGFP-TBX4B cells were consistent with those of sorted Tax⁺ TBX4B cells. We conclude that the lower frequency of chromatin looping observed in Tax⁺ cells was associated with proviral transcription and was not caused by staining Tax protein. Also, the observation of similar changes in looping frequency in both the transduced and non-transduced cells indicates that transduction *per se* did not affect the results of q4C analysis.

A decrease in the number of q4C peaks at distant chromatin contacts in the host genome was observed in provirus plus-strand-expressing subsets in all T cell clones investigated (**Figs 2B, S2 and S3**), in regions either with or without CTCF binding sites (**Fig 2C**).

To corroborate these observations by an independent technique, 3C-qPCR was performed to quantify the frequency of the interaction between the provirus and the two host regions where the principal 4C peaks were identified in the Tax⁻population of clone 11.63 (**S3 Fig**),

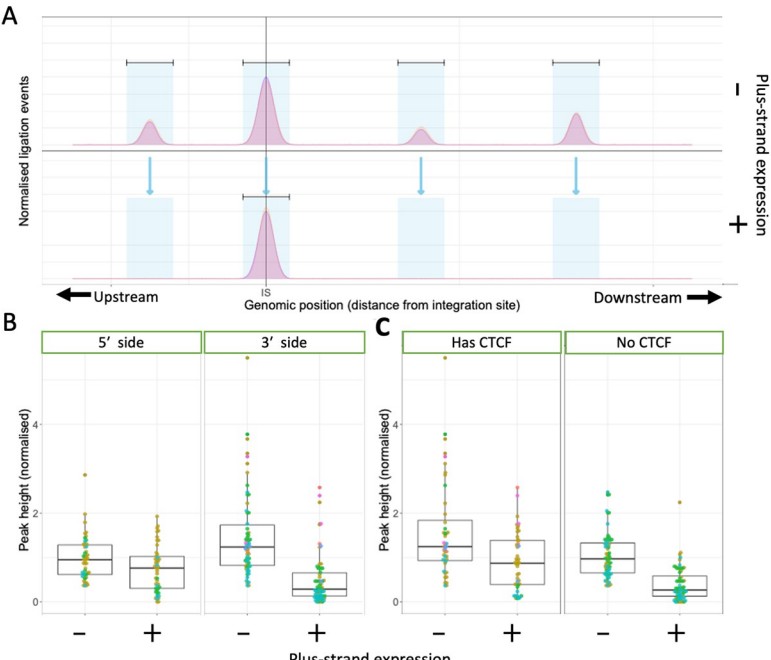

**Fig 3. HTLV-1 plus-strand expression results in a significant reduction in contact frequency (q4C peak height) with the host genome (A)** Schematic to show quantification of change in peak frequency with plus-strand expression. First, reproducible q4C peaks were identified in the non-expressing (Tax-negative) cells, as previously described [10]. Then, peaks were sought (using the same algorithm) in each corresponding genomic location in the plus-strand-expressing cell population. **(B)** Normalized peak height of q4C peaks identified in all clones analysed, respectively on the 5′ and 3′ sides of the provirus. Peak height is defined as the maximum number of ligation events per region (normalised to total ligation events in a sample) calculated for each peak region. p = 0.027 and p = 9.7 * 10$^{-13}$ for upstream and downstream regions, respectively (unpaired two-tailed Wilcoxon test. **(C)** Normalized peak height of q4C peaks identified in all clones analysed, comparing peaks that contain a CTCF site and those without CTCF sites. Peak height was significantly greater in non-expressing cells, both in peaks with a CTCF site and those without (p = 0.0018 and p = 1.8 * 10$^{-13}$, respectively, unpaired two-tailed Wilcoxon test).

using specific primers designed to amplify the junction between the provirus and the host genome (**S4A Fig and S1 Table**). The results showed a reproducible significant decrease in chromatin contact frequency between the provirus and both peaks in Tax$^{+}$ cells (**S4C and S4D Fig**), although there was no difference in the contact frequency observed within the provirus (**S4B Fig**).

Next, to quantify the difference in chromatin loop formation between provirus-expressing and non-expressing cells, peak regions were defined in the non-expressing cell population, and peaks were then sought at the same sites in the provirus-expressing cell population. (**Fig 3A**) At each site, we then compared the normalized peak height of the q4C profile in non-expressing cells with that in provirus-expressing cells. The results show that HTLV-1 plus-strand expression resulted in a significant reduction in the frequency of these chromatin loops, especially on the 3′ side of the integration site (**Fig 3B**). This reduction was observed in both regions with and without CTCF binding sites (**Fig 3C**).

## HTLV-1 proviral plus-strand expression drives host transcription downstream *in cis*

Throughout this work, we refer to the direction of flanking host transcription relative to the orientation of the provirus. Thus, host transcription upstream (5′) of the 5′LTR, in the same

sense as the proviral plus-strand, is denoted as 'same-sense, 5′ side of provirus' (**Fig 1 arrow a**), and transcription in the same sense 3′ of the 3′LTR as 'same-sense, 3′ side of provirus' (**Fig 1 arrow b**). Similarly, we refer to host transcription in the same sense as the proviral minus-strand as 'antisense, 3′ side of provirus' (**Fig 1 arrow c**) and 'antisense, 5′ side of provirus' (**Fig 1 arrow d**).

Next, we asked whether the aberrant host transcription is associated with proviral transcription or simply with the presence of the provirus, regardless of transcription. The landscape of host transcription flanking the provirus has been investigated in ATL cases [19,20], in which there is usually a dominant single malignant clone. However, there are typically widespread epigenetic and transcriptional abnormalities in malignant cells, and expression of the proviral plus-strand is frequently lost in ATL clones [21]. We previously reported abnormal *in cis* host transcription near the integration site in naturally-infected, non-malignant HTLV-1-positive T-cell clones [10], but since only a fraction of cells express the proviral plus-strand at a given instant, we could not distinguish whether the abnormal host transcription was associated with proviral plus-strand expression, or whether there was constitutive activation irrespective of the plus-strand burst, perhaps by the proviral enhancer.

RNA isolated from fresh, unfixed cells is required for optimal RNA-seq results. However, intracellular staining of Tax protein requires prior fixation. Therefore, to elucidate whether the aberrant host transcription was induced by proviral plus-strand transcription, we used d2EGFP-transduced clones (11.63, 11.50 and TBX4B) (see Table 1) [11], from which GFP⁺ (Tax-expressing) and GFP⁻ (Tax-non-expressing) cells were isolated by live-cell sorting; total cellular RNA was extracted and subjected to a stranded RNA-seq analysis. The proviral plus-strand was highly transcribed in all 3 clones in GFP⁺ cells but not in GFP⁻ cells (**Fig 4A and 4B**), confirming that GFP⁺ cell-sorting enriched a cell population with active HTLV-1 plus-strand transcription.

Host transcription in the same sense on the 3′ side of the provirus near the clone's respective proviral integration site was greater in GFP⁺ cells than in GFP⁻ cells in all 3 clones (**Fig 4C**). For example, in clone d2EGFP-TBX4B, HTLV-1 is integrated between exon 2 and exon 3 of the *PNPLA3* gene (**Fig 4D**), which is not normally expressed in T cells (**Fig 4C**). The *PNPLA3* gene (same-sense, 3′ side of the provirus) was highly transcribed in the GFP⁺ population, but was not transcribed in GFP⁻ cells of clone d2EGFP-TBX4B, or in two other clones (d2EGFP-11.50 and d2EGFP-11.63) regardless of Tax expression, suggesting that the observed increase in *PNPLA3* mRNA did not result from Tax-mediated transactivation (**Fig 4C**). Aberrant transcription in the same-sense on the 5′ side of the provirus was also seen in two clones (11.50 and TBX4B) at a low level (**Fig 4C and 4D**).

In addition to splicing between the 3′LTR and the host genome (antisense, 5′ side of provirus), which has also been reported in ATL cases [19], in plus-strand-expressing cells we found that the exons of 5′LTR (exon H1/exon H2 in **Fig 1**) were spliced out to form not only normal viral mRNAs but also viral-host fusion transcripts, by fusion to downstream host plus-strand transcripts. For example, in the GFP⁺ subset of d2EGFP-TBX4B cells, proviral exons H1 or H2 were found to be fused to exon 3 of *PNPLA3* (**Fig 4D and 4E**). The observed events always occurred between a canonical HTLV-1 splice donor and a canonical host splice acceptor. Such splicing events were observed even when no host gene was present in the flanking region (**S5 Fig**). We then reanalysed the RNA-seq data obtained previously [11] and found a novel transcript in clone Timer-3.60 cells expressing the plus-strand, formed by splicing out of the proviral plus-strand exon H1 and fusion to novel host exons (**S5B Fig**). These splicing events again followed the canonical GT|AG mRNA processing rule. Fusion was observed between a host splice acceptor and proviral splice donor up to 120 kb away. By contrast, splicing was not observed between a host splice donor and an HTLV-1 splice acceptor; although low-level host

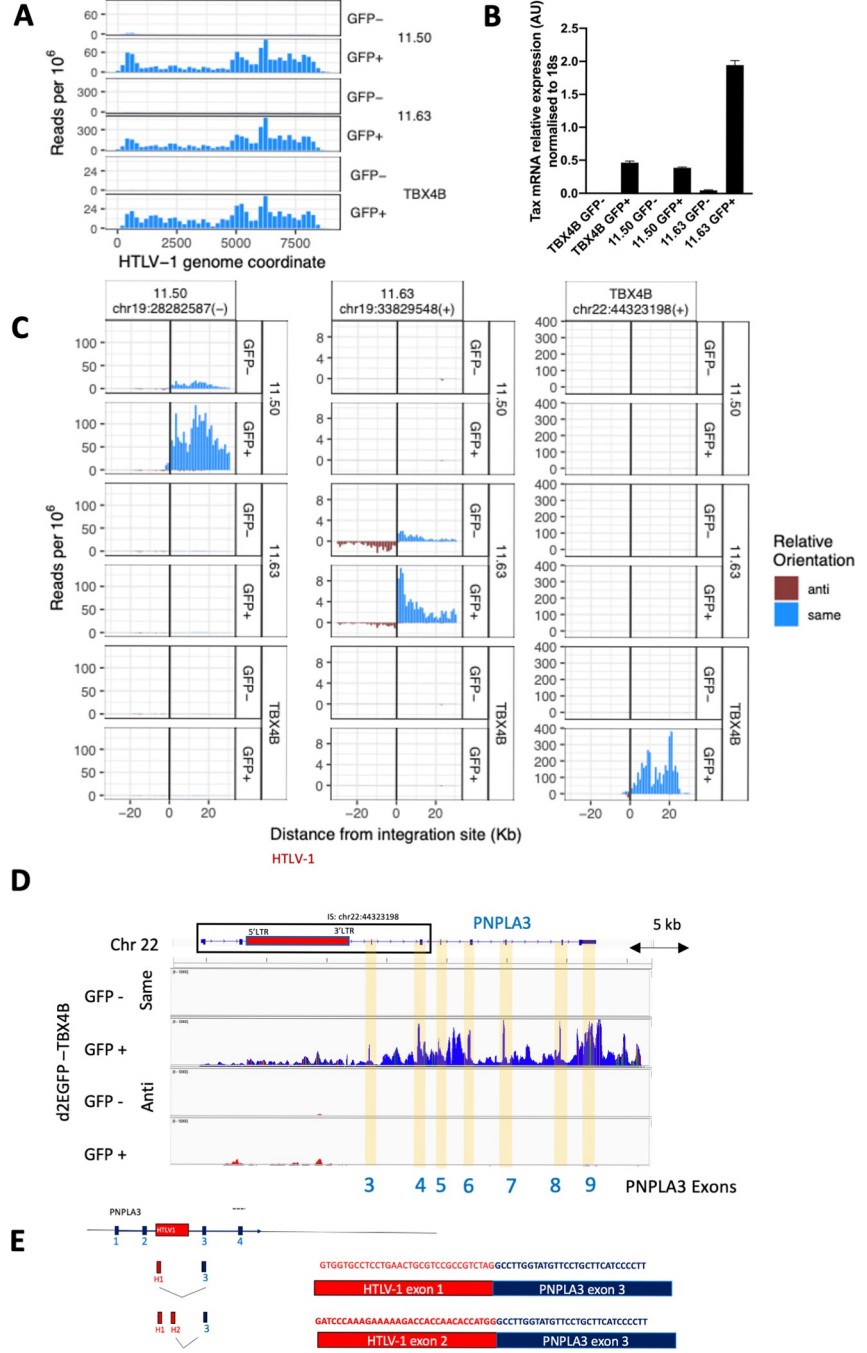

**Fig 4. Proviral and host transcription and splicing in live-sorted T cell clones.** (A) RNA-seq analysis of HTLV-1 proviral expression in live-sorted d2EGFP clones (TBX4B, 11.50 and 11.63). (B) Tax expression measured by qPCR with primers specific for *tax* mRNA or *18S* ribosomal RNA (18S rRNA). qPCR plots are expression values normalized to 18S rRNA. Data represent a mean of two biological replicates; error bars are SEM. AU—arbitrary units. (C) Host RNA expression 30kb on either side of the proviral integration site. On the horizontal axis, positive values denote positions extending from the 3′ LTR side of the provirus; negative values denote positions 5′ of the 5′LTR. Each row shows the transcription density (normalized RNA-seq read count) flanking that genomic position in the clone indicated at the right-hand side. In each case, transcription orientation and positions are shown relative to the integrated provirus. Read density shown in blue shows transcription in the same orientation as the proviral plus-strand (same sense); red shows transcription in the opposite sense to the proviral plus-strand (antisense). (D) Identification of splice sites of viral-host fusion transcripts in d2EGFP-TBX4B clone cells. Coverage tracks of same sense transcription (blue) and antisense transcription (red) in Integrative Genomics Viewer (IGV). Exons of PNPLA3 in the 3′ side of the

integration site are highlighted in yellow. (E) Fusion transcripts between an HTLV-1 plus-strand major splice donor (red, proviral exon H1 or H2) and the canonical splice acceptor site in the host *PNPLA3* gene (blue, *PNPLA3* exon 3) were identified in GFP$^+$ (HTLV-1 plus-strand-expressing) cells. To identify splice sites of fusion transcripts, reads were aligned to a reference genome (hg19) containing the HTLV-1 provirus (AB513134) genome in the TBX4B clone integration site at chr22:44323198. Fusion transcripts are shown with fused sequences.

transcription in the same-sense on the 5′ side of the provirus was induced by proviral plus-strand expression in two clones (**Fig 4C and 4D**).

## Treatment of T cell clones with an inhibitor of transcriptional elongation allows recovery of chromatin loops in the Tax$^+$ population

The observation of upregulated host transcription near the integration site and decreased chromatin loops in the provirus-expressing cell populations raised the question whether provirus-induced host transcription disrupts chromatin loop formation between the provirus and host genome. To test this hypothesis, we treated the infected cells with flavopiridol, an inhibitor of transcriptional elongation. Flavopiridol was reported to inhibit both elongation and readthrough transcription induced by influenza virus [22]. Treatment of cells with 1 nM flavopiridol for 1.5 hours did not affect Tax protein expression (S6 Fig) so the cells could still be sorted for Tax protein. Flavopiridol treatment also did not reduce *tax* mRNA expression significantly. However, the level of aberrant host transcription in the same sense, 3′ side of the provirus was significantly decreased (**Fig 5A and 5B**).

After flavopiridol treatment, the cells were stained intracellularly, sorted for Tax protein, and then subjected to q4C assay. While the frequency of chromatin loops between the provirus and the host genome was lower in Tax$^+$ cells than in Tax$^-$ cells (as in Fig 1 above), chromatin loops remained in Tax$^+$ cells when they were treated with flavopiridol (**Fig 5C**). We conclude that inhibition of transcription elongation by flavopiridol reduced host transcription and allowed the preservation and restoration of chromatin loops.

## Discussion

The results presented here lead to two main conclusions. First, q4C analysis of sorted T cell clones for plus-strand expression revealed that the chromatin loops formed between the provirus and the immediately flanking host genome are reduced or lost during plus-strand proviral transcription. Second, plus-strand proviral transcription causes (1) aberrant host transcription on the same-strand, chiefly on the 3′ side of the provirus and to a lesser extent on the 5′ side; (2) fusion transcripts between provirus and host RNA; the 5′LTR (exon H1/exon H2) was fused to host exons in the same sense on the 3′ side of the provirus, although intra–HTLV-1 splicing remained intact.

In clone TBX4B we observed that transcriptional activation of a host gene more distant (1.4 Mb) from the provirus was associated with expression of the wild-type provirus, but not with the provirus in which the CTCF binding site was knocked out (S7 Fig). We postulate that, while the nearby chromatin loops are diminished or lost during HTLV-1 proviral plus-strand transcription, more long-range CTCF-dependent loops can remain intact. This result (S7 Fig) suggests that the long-range enhancer effect of HTLV-1 requires the CTCF binding to the provirus and active viral transcription. Further investigation will be needed to clarify this mechanism.

The chromatin loops formed by the provirus reported here and previously [10] are made with adjacent regions of the host genome. The frequency of these loops declines rapidly within ~5 Mb of the provirus. These observations are consistent with the size distribution of normal

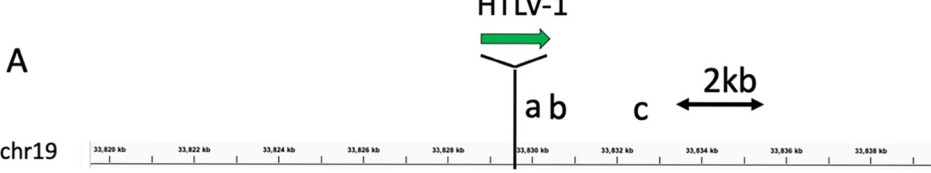

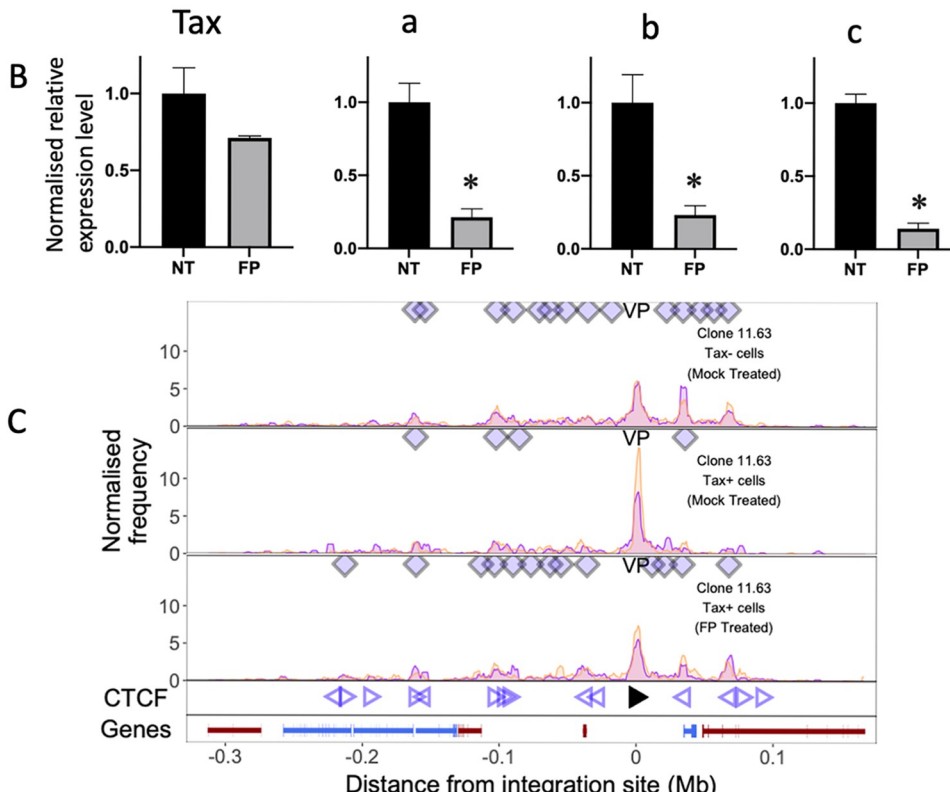

**Fig 5. Treatment of T cell clones with a transcriptional elongation inhibitor allows recovery of chromatin loop formation in the Tax+ population.** (A) After treatment with 1 nM flavopiridol (FP) for 1.5 hrs, total RNA was extracted from clone 11.63 and subjected to RT-qPCR for *tax* and three regions in the 3′ flanking host genome, respectively a: +188 bp, b: +535 bp and c: +3,198 bp from the 3′ end of the provirus. (B) Relative expression intensity (normalized to 18s rRNA) of *tax* and the host genome at positions a, b and c. Data are mean ± SEM. (N = 3). * $P<0.05$ (paired t-test). (C) q4C profiles of mock-treated Tax- cells (top track) and Tax+ cells (middle track), and flavopiridol (FP)-treated Tax+ cells (bottom track). Diamonds mark the positions of reproducible chromatin contact sites identified by the peak calling algorithm. Open arrowheads denote positions of CTCF-BS; the filled arrowhead denotes the CTCF-BS in the provirus. Gene track shows RefSeq protein-coding genes in the flanking host genome.

chromatin loops in the human genome [23], and with the mechanism of loop extrusion by which these loops are formed [24]. It is known that contacts can also be made with more distant locations on the genome, including other chromosomes; however, such distant contacts are typically much less frequent, although certain trans-chromosomal contacts may be evolutionarily conserved [25].

Aberrant read-through transcription and splicing is a known feature of retroviral transcription [26]. Previous studies reporting RNA-seq analysis in ATL cases [19,20] showed aberrant antisense expression in the host genome flanking the 5′ side of the provirus; the 3′LTR sequence was spliced out and fused to host exons on the antisense strand on the 5′ side of the

provirus. However, aberrant host transcription in the same sense as the proviral plus-strand (either 5′ or 3′ to the provirus), and splicing between the 5′LTR and host exons in the same sense on the 3′ side of the provirus, were rarely observed. In the non-malignant clones in the present study, although viral-host transcripts in the minus-strand were sometimes observed, plus-strand transcription always dominated.

We observed splicing between the 5′LTR and host exons in the plus-strand on the 3′ side of the provirus (**Figs 4D, 4E and S5**), even when there was no identified host gene in either strand. The observation that proviral-host chimeric RNA splicing events can activate cryptic host exons was recently reported in HIV-1-infected Jurkat T cell clones carrying reporter proviruses integrated into the introns of three cancer-related genes [27]. Our observations show that HTLV-1 plus-strand expression has the potential to disrupt host cell functions by creating novel transcripts, with potential biological functions, under the transcriptional control of the HTLV-1 5′LTR. To explore this possibility, further research is needed to test the stability and activity of these aberrant mRNAs.

Whereas frequent splicing was observed between a proviral donor and host acceptor, we did not observe splicing between a host donor and proviral acceptor, even if the host genome 5′ side region was transcribed. This observation implies that any same-sense host transcription on the 5′ side of the provirus stops at the integration site, whereas LTR-driven transcription can result in read-through into the host genome in the same sense on the 3′ side of the provirus.

CTCF, which binds to the HTLV-1 provirus, can regulate mRNA splicing: further work is required to test the possibility that CTCF bound to the provirus causes aberrant splicing in the flanking host genes.

The results reported here show that the aberrant host transcription induced by the HTLV-1 provirus differs from the host gene transcriptional landscape at the HIV-1 integration site [27]. No aberrant host transcription was observed in the antisense strand in HIV-1-infected cells. By contrast with HIV-1, HTLV-1 can induce host transcription in the same sense on the 5′ side of the provirus (**Fig 4C**), whereas both proviruses induce strong same-sense transcription on the 3′ side of the provirus. Whereas downstream transcription can be at least partially explained as readthrough transcripts from the provirus, same-sense transcription on the 5′ side of the provirus in an HTLV-1[+] clone cannot be explained by readthrough from the 5′LTR. Two observations suggest that this same-sense transcription on the 5′ side of the provirus is due to an enhancer effect of the HTLV-1 5′LTR. First, the aberrant same-sense transcription was seen only in GFP[+] cells (**Fig 4C and 4D**); second, the transcription correlated with the level of plus-strand proviral transcription in timer protein reporter clones (S8 Fig).

An unexpected finding in this study was that aberrant loop formation between the provirus and host chromatin is suppressed by proviral plus-strand transcription. Since CTCF appears to remain bound to the provirus during expression of *tax* [8], the observed decrease in chromatin looping cannot be attributed to loss of CTCF binding. The observations that proviral transcription increases aberrant transcription and splicing in the flanking host genome (**Figs 4 and S5**) and that reducing proviral-expression-induced host transcription with an inhibitor of transcriptional elongation allows recovery of chromatin loops in the Tax[+] population (**Fig 5**) suggest that aberrant host transcription induced by the proviral expression disrupts chromatin looping. This inference is consistent with the report by Heinz et al. (2018) [22] that during influenza A virus (IAV) infection the IAV non-structural protein 1 (NS1) induces global inhibition of transcription termination of highly transcribed genes and causes readthrough transcription for hundreds of kilobases, resulting in disruption of chromatin interactions.

Chromatin looping between the HTLV-1 provirus and the host genome is mediated by CTCF binding to the provirus [14]. We postulated that CTCF binding confers a selective

advantage on the provirus; however, the mechanism of this putative advantage is not known. At least three non-mutually-exclusive possibilities can be identified. First, through its function as an epigenetic barrier, CTCF may regulate epigenetic modification of the provirus [28] Second, CTCF (again through its barrier function) might block unwanted activation of plus-strand proviral transcription by the recently reported intragenic enhancer [29]. Third, CTCF may control the intranuclear position of the chromatin containing the provirus, which is correlated with selective survival of HTLV-1-infected T cell clones in vivo [30].

## Methods

### Ethics statement

All donors gave written informed consent in accordance with the Declaration of Helsinki to donate blood samples to the Communicable Diseases Research Tissue Bank, approved by the UK National Research Ethics Service (15/SC/0089).

### Cells

The details of HTLV-1-infected T-lymphocyte clones used in this study are shown in Table 1. All clones were derived as previously described [11] from peripheral blood mononuclear cells (PBMCs) of donors attending the National Centre for Human Retrovirology (NCHR) at Imperial College Healthcare NHS Trust, St Mary's Hospital, London. The identification of genomic insertion sites by LMPCR was described elsewhere [31]. The cells were maintained in RPMI-1640 (Sigma, R0883) supplemented with L-glutamine, penicillin+streptomycin and 20% fetal bovine serum (Gibco, 10500–064) in 5% $CO_2$ at 37˚C. IL-2 (Miltenyi Biotec, 130-097-745) was supplemented (100 unit/ml) into the culture twice a week. The integrase inhibitor raltegravir (Selleck Chemicals, MK-0518) was used at 10 μM throughout the culture to prevent secondary infection.

### 3C and q4C assay of sorted cells

HTLV-1- infected T cell clones were stained with LIVE/DEAD Fixable Near-IR Dead Cell Stain kit (Invitrogen, L34976) to enabling gating on live cells and then crosslinked in phosphate-buffered saline (PBS) containing 1% formaldehyde for 10 min at room temperature. Then the Tax protein was stained intracellularly with anti-Tax-AF647 (clone LT-4; 0.4 μg/ml), using a Foxp3 staining kit (eBioscience, 00-5523-00) and viable Tax$^+$ and Tax- cells were sorted (**S1 Fig**). For the Tax expression reporter clones (d2EGFP clones), the cells were stained with LIVE/DEAD Fixable Near-IR Dead Cell Stain kit, crosslinked as above, and viable GFP$^+$ cells and GFP- cells were sorted. After sorting, q4C was performed [10].

q4C analysis was carried out as previously described [10]. Reads spanning 100 to 150 bp were inspected for quality using FastQC, and filtered for correct amplification using the NlaIII restriction sequence and preceding 4 bases (total 8 bases) as an identifying barcode for the q4C viewpoint using Cutadapt. Reads where the first NlaIII site was incompletely digested (therefore containing an additional NlaIII fragment) were further trimmed by Cutadapt. Finally, Trimgalore was used to trim low quality reads (<20) and remove the Ilumina adaptor sequence, keeping reads of minimum 30 b in length. Trimmed reads were aligned to a combined reference of human (hg19) and viral (AB513134) genomes using Bowtie2. Ligation sites were quantified from aligned reads using Perl and R scripts. Ligation sites were quantified in windows of 10 kb across the chromosomes in order of base coordinates. Peaks were called as described previously [10] using a three-state hidden Markov model, requiring agreement between both samples (biological replicates).

Real-time 3C qPCR was described elsewhere [17]. Primer pairs and probe (**S1 Table**) were used to detect long-range chromatin contacts between the provirus and host genome region at Peak 1 (34.5 kb downstream of the provirus) and Peak 2 (68 kb downstream of the provirus) of chromosome 19. As control, we used a primer pair and Taqman probe to detect the contacts of two regions in the provirus, and a 101 bp sequence from the provirus was amplified to normalize the qPCR using internal control primer sets and probe (**S1 Table**). Data analysis was done with LinRegPCR software (version 2014.5). Six replicates were performed on each library, using a QuantStudio 7 Flex Real-Time PCR System utilizing TaqMan Gene Expression Master Mix (Applied Biosystems).

## RNA-seq and RT-qPCR

d2EGFP clones were stained with LIVE/DEAD, and live cells were sorted with a BD FACSAria lll cell sorter under containment level 3 (CL3) conditions, as described elsewhere [11]. Total RNA was extracted from the sorted cells using an miRNeasy kit (Qiagen) and RNA-seq libraries were prepared using the Ribo-Zero Plus rRNA Depletion kit and TruSeq Stranded mRNA HT Sample Prep Kit and sequenced with the NovaSeq6000 (150 bp paired-end reads).

RNA-seq reads were inspected for quality using FastQC, and trimmed using Trimgalore to remove low-quality reads and Ilumina adapter sequences. Trimmed reads were aligned against a combined reference of human (hg19) and viral (AB513134) genomes using GSNAP v. 2019-06-10. Read coverage was counted using Bedtools coverage after split to exons using Bedtools bamtobed tool using the resolution denoted in each figure.

Reads were aligned to a combined reference of human (hg19) and viral (AB513134) inserted into the integration site at chr22:44323198, chr19:33829548 and chr04:70567285 to identify splice sites of fusion transcripts in TBX4B, 11.63 and 3.60, respectively.

RNA-seq of timer protein reporter clones was described elsewhere [11]. RNA-seq of TBX4B clones containing wild-type CTCF Binding site (BS) in the provirus and mutant (Mut) clone containing a mutated CTCF-BS were performed as described elsewhere [10].

For RT-qPCR, first-strand cDNA was synthesized with First Strand cDNA Synthesis Kit using random primers. Primers used are shown in **S1 Table**. Real-time qPCR was then performed in 6 replicates on each library, using a QuantStudio 7 Flex Real-Time PCR System utilizing Fast SYBR Green Master Mix (Applied Biosystems).

## Supporting information

**S1 Table. DNA sequence of primers and probes used in 3C-qPCR and RT-qPCR.**
(TIFF)

**S1 Fig. Flow sorting of Tax⁻ and Tax⁺ cells.** HTLV-1-infected T cell clone 11.63 was stained for live cells, crosslinked in 1% formaldehyde, stained intracellularly for Tax protein, and flow-sorted to isolate Tax⁻ and Tax⁺ subsets (see Methods).
(TIFF)

**S2 Fig. Flow sorting of provirus-expressing and non-expressing cells by an independent technique produced similar q4C profiles.** (A) q4C profiles of Tax⁻ (upper panel) and Tax⁺ (lower panel) cells sorted from clone TBX4B after intracellular staining of Tax. (B) q4C profile of non-expressing (GFP⁻) (upper panel) and provirus-expressing (GFP⁺) (lower panel) cells isolated from d2EGFP-TBX4B clones, selected by GFP signal (without Tax staining).
(TIFF)

**S3 Fig. Transcription of the proviral plus-strand is accompanied by a loss of contacts between the provirus and flanking host chromatin.** (A) q4C profiles of Tax⁻ (upper panel)

and Tax$^+$ (lower panel) cells from clone 11.63. (B) q4C profiles of non-expressing (GFP$^-$) (upper panel) and provirus-expressing (GFP$^+$) cells (lower panel) from clone d2EGFP-11.50. Vertical axes show the normalized frequency of chromatin contacts between the provirus and the host genome.
(TIFF)

**S4 Fig. Quantitative 3C (3C-qPCR) analysis confirmed decreased frequencies of chromatin looping. (A)** q4C profile of Tax$^-$cells of clone 11.63. The technical peak seen in the q4C view-point (VP) and two of the main peaks (Peak 1) and (Peak 2) identified in the output of the Tax$^-$fraction of clone 11.63. (B) As control, the frequency of chromatin interactions was quantified by 3C-qPCR on sorted Tax$^-$and Tax$^+$ cells, using a primer pair and Taqman probe to detect the contacts between two regions: VP and another region in the provirus (**S1 Table**). There was no significant difference between Tax$^+$ and Tax$^-$cells (combined p value = 0.932, Fisher's method of combining p values). (C and D) Primer pairs and probe were used to detect long-range chromatin contacts between the provirus and host genome region at Peak 1 (C) or Peak 2 (D). Results of 3C-qPCR of two biological replicates (rep) are shown. Peak1 contact frequency was significantly higher in Tax$^-$cells than in Tax$^+$ cells (combined p value 0.012, Fisher's method). Peak2 contact frequency was significantly higher in Tax- cells than in Tax$^+$ cells (combined p value 0.000607, Fisher's method).
(TIFF)

**S5 Fig. Fusion transcripts between HTLV-1 provirus and host genome.** (A) Identification of splice sites of fusion transcripts in the plus-strand-expressing cells of clone d2EGFP-11.63 and (B) in timer protein reporter clone Timer-3.60. Plus-strand fusion transcripts between HTLV-1 exon1 (H1) and same sense, 3′ side host gene exon (blue) or novel host exons (green) are shown with fused sequences.
(TIFF)

**S6 Fig. Treatment with flavopiridol for 1.5 hrs did not alter Tax protein expression.** Clone 11.63 cells were treated with 1nM flavopiridol for 1.5 hrs and then stained for Live/Dead and then Tax protein.
(TIFF)

**S7 Fig. Expression of distant host genes correlates with expression of *tax*.** (A) Normalized mRNA read counts of two genes that lie >1.4 Mb from the provirus in clone Timer-TBX4B, in the four successive phases of the HTLV-1 plus-strand transcriptional burst: DN–double negative (HTLV-1 silent); blue–early burst; DP double-positive (mid-burst); red–late burst. Results of two independent experiments are shown. Expression of both *SMC1B* and *RIBC2* closely followed the trajectory of the HTLV-1 burst in clone Timer-TBX4B, but not in the unrelated HTLV-1-infected clone Timer-3.60. Data from [11]. (B) Knockout of the CTCF binding site in the provirus in clone Timer-TBX4B (middle panel) abolished the transcription of both *SMC1B* and *RIBC2* observed in the wild-type clone (lower panel). Results of two independent experiments are shown. Neither gene was expressed in an unrelated HTLV-1-infected clone ED. The results suggest that maintenance of a CTCF-dependent chromatin loop between the host genome and the provirus is required for the burst of transcription of these distant genes associated with the HTLV-1 plus-strand burst.
(TIFF)

**S8 Fig. RNA-seq analysis of HTLV-1 proviral expression in live-sorted cells of clone Timer-TBX4B.** Cells were sorted into four populations based on the fluorescence of the Timer protein, DN–double negative (HTLV-1 silent); blue–early burst; DP double-positive (mid-

burst); red–late burst. [11].(A) Coverage tracks in IGV of plus strand HTLV-1 provirus transcription and (B) host gene PNPLA3 transcription. (C) Transcription in *PNPLA3* exons 1 and 2 (note range on vertical axis 0 to 500).
(TIFF)

## Acknowledgments

We thank Parisa Amjadi from the CL3 Cell Sorting Facility at The Centre for Immunology and Vaccinology at Imperial College London for cell sorting. We are grateful to Tomas Fitzgerald at the European Bioinformatics Institute and Aris Aristodemou for helpful discussion. We thank Laurence Game at MRC London Institute of Medical Sciences for q4C sequencing, and Oxford Genomics Centre for cDNA library preparation and RNA sequencing.

## Author Contributions

**Conceptualization:** Hiroko Yaguchi, Charles R. M. Bangham.

**Data curation:** Anat Melamed.

**Formal analysis:** Anat Melamed.

**Investigation:** Hiroko Yaguchi, Saumya Ramanayake, Helen Kiik, Aviva Witkover.

**Methodology:** Hiroko Yaguchi, Anat Melamed, Saumya Ramanayake, Helen Kiik, Aviva Witkover.

**Project administration:** Charles R. M. Bangham.

**Resources:** Charles R. M. Bangham.

**Software:** Anat Melamed.

**Supervision:** Charles R. M. Bangham.

**Writing – original draft:** Hiroko Yaguchi, Charles R. M. Bangham.

**Writing – review & editing:** Hiroko Yaguchi, Anat Melamed, Saumya Ramanayake, Helen Kiik, Aviva Witkover, Charles R. M. Bangham.

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
