## [Decision Letter · Decision Letter 0]

1 Nov 2023

Dear Dr. Bangham,

Thank you very much for submitting your manuscript "The impact of HTLV-1 expression on the 3D structure and expression of host chromatin" for consideration at PLOS Pathogens. As with all papers reviewed by the journal, your manuscript was reviewed by members of the editorial board and by three independent reviewers. In light of the reviews (below this email), we would like to invite the resubmission of a significantly-revised version that takes into account the reviewers' comments.

We cannot make any decision about publication until we have seen the revised manuscript and your response to the reviewers' comments. Your revised manuscript is also likely to be sent to reviewers for further evaluation.

Sincerely,

Edward William Harhaj, Ph.D.

Academic Editor

PLOS Pathogens

Susan Ross

Section Editor

PLOS Pathogens

Kasturi Haldar

Editor-in-Chief

PLOS Pathogens

orcid.org/0000-0001-5065-158X

Michael Malim

Editor-in-Chief

PLOS Pathogens

orcid.org/0000-0002-7699-2064

Reviewer's Responses to Questions

**Part I - Summary**

Reviewer #1: The manuscript by Yaguchi et al describes studies of 3D conformation analysis and HTLV-1 transcription in infected cell lines and reported that plus strand transcription reduces aberrant chromatin loops with the CTCF binding site in the integrated provirus. The results are not interpretable in light of the fact that some of data (e.g. Fig 2) were not printed properly. In addition, key controls are missing. The functional significance of the chromatin loops with regards to infected cellular transcription, proliferation, or survival were not examined. Lastly, the manuscript is very difficult to read.

Reviewer #2: Previously, this group reported that the HTLV-1 genome contains a CTCF binding site involved in creating abnormal chromatin looping with the host cell genome. In the current manuscript, Yaguchi et al. seeks to analyze the effect of HTLV-1 transcription on chromatin architecture and transcription of neighboring DNA. From several HTLV-1-infected clones, the authors sorted the cells into populations based on the presence or absence of Tax expression (i.e. sense transcription). Using chromosome conformation capture (q4C and 3C) they observed that the provirus sense transcription reduces chromatin loops with the provirus, regardless of the presence of CTCF sites in the host genomic regions. In addition, correlating with the loss of chromatin loops, sense transcription is linked to increased transcription mainly from the 3’ side of the provirus, probably due to read-through transcription. This is an interesting study that confirms the profound effect of HTLV-1 integration and transcription on the host cell chromatin architecture and host transcription. A few comments below will help to clarify the manuscript and the findings.

Reviewer #3: HTLV-1 proviral DNA contains a CTCF binding site thought to be involved in regulating the formation of chromatin loops. This manuscript by Yaguchi et al. examined the impact of the HTLV-1 provirus on the structure and transcriptional activities of the host DNA sequences nearby. HTLV-1-infected cells undergoing Tax-mediated viral transcription or not were isolated and analyzed by a technique known as the quantitative chromosome conformation capture (q4C) and RNAseq. The study suggests that Tax-mediated proviral plus strand transcription alters the mRNA transcription and splicing of the host exons downstream of the viral 3’ LTR and reduces the formation of chromatin loops. The CTCF site in HTLV-1 proviral DNA appears to affect the transcriptional activation of a host gene distal to the proviral DNA.

Overall, the authors employed state-of-the-art techniques in the study. The experiments were well-controlled, and the data obtained were convincing. The most likely explanation for the “aberrant” host mRNA transcription observed is the transcriptional readthrough, initiating at the 5’ LTR (driven by Tax), elongating across 3’ LTR into the host gene, and followed by mRNA splicing with the HTLV-1 1st exon spliced with the host exon immediately downstream of the 3’ LTR. The reduction in chromatin loop formation during active transcription is novel and interesting. The notion that chromatin loop formation may allow the HTLV-1 provirus to activate distal genes (via Tax-bound HTLV-1 enhancer?) is hinted at but not fully developed. The authors can improve the manuscript further by addressing the following comments.

**Part II – Major Issues: Key Experiments Required for Acceptance**

Reviewer #1: Major Concerns

1) The graphs in Fig 2A are missing. The numbers of peaks listed in Fig 2B are very small and inadequate for any firm conclusions.

2) Almost all of the results are from subjects with HAM/TSP. Are these results unique to this subset?

3) It is unclear whether or not fixation, flow sorting, or transduction of reporters affected the chromatin conformation analysis or transcription results. Studies should have been performed to examine chromatin loops and transcription not involving the provirus to show that there is no effect of Tax expression.

4) Why does transcription affect chromatin contacts outside the provirus but not within the provirus? This seems counterintuitive.

5) The changes in 4C profile in Fig 5C with flavoperidol are subtle and limited to a single cell line, and thus not convincing. Why did flavoiridol not decrease Tax mRNA levels?

Reviewer #2: 1) Figure 3 is missing some information. Are we looking at all the q4C peak regions of all cell clones analyzed? The genomic position (distance from integration site) is missing numbers (Mb) to denote the positions analyzed.

2) Figure 5 does not show a reduction of HTLV-1 sense transcription correlating with a reduction of 3’ host sense transcription, since the elongation inhibitor does not significantly affect Tax mRNA. The authors should explain or speculate on reasons for this discrepancy (e.g., Tax (HTLV-1) mRNA stability?). Alternative experiments should be developed to better correlates HTLV sense transcription with read-through transcription and chromatin loops. Possibly, the authors could show RNA polymerase II occupancy (phosphorylation status?) at the HTLV-1 provirus before and after flavopiridol treatment.

3) The authors show by q4C that the sense transcription affects close and long-range chromatin loops; however, they focus on transcription in closed proximity of the provirus. While they show that provirus read-through sense transcription affects transcription in the neighboring host genome, they do not show the effect of sense transcription/chromatin changes on transcription at distal host genomic sites, and therefore, the study appears incomplete. The study would be made more thorough by determining the status of transcription of the genes located further downstream of the HTLV-1 provirus. Specifically, the authors could analyze genes further downstream of regions a, b and c in Figure 5, clone 11.63. If the provirus transcription affects long-range chromatin loops, one would think that the transcription of genes within these loops would also be affected. It seems that S7 Fig in the Discussion section is an attempt to demonstrate long-range effects, but without much explanation, this figure is difficult to interpret. Also, the data were derived from a different experimental design.

4) Integrating S7 Fig and S8 Fig in the Discussion without much explanation does not make the reading and comprehension of the manuscript easy. The manuscript would benefit from having these figures described in more detail at appropriate points in Results section.

Reviewer #3: 1. The chromatin loops formed by the proviral DNA involve only host DNA sequences that are adjacent to the integration sites and syntenic. Is this because of the constraint of the q4C technique or the nature of chromatin loops in general? The authors should discuss these points in more depth.

2. It is reported that there are between 15000 and 40000 CTCF binding sites in the human genome. Do most chromatin regions that HTLV-1 proviral DNA interacts with (as mapped in this study) contain one or more CTCF sites?

3. It should be pointed out to the readers that the so-called aberrant host transcription and splicing observed is primarily due to mRNA transcriptional readthrough and splicing (Fig. 4, Fig. S5). This is well known for retroviral transcription, and HTLV-1 is no exception.

4. The TBX4B study (Fig. S7) suggests the long-range “enhancer” effect of HTLV-1 requires the CTCF site and active viral transcription (with the recruitment of Tax-CREB-CBP/p300?). Is this observed in other HTLV-1-infected cell clones in this study?

**Part III – Minor Issues: Editorial and Data Presentation Modifications**

Reviewer #1: Minor Concerns

6) In Table 1, if the designations of subjects are patient initials they should be removed. Also, it is not clear to what the provirus orientation is compared – the centrosome?

7) Table 2 should be combined with Table 1. Why did they not look at Tax expression in all the cell clones? Why were so many cells positive in 3 cell lines (36-56% of cells) when the text says that “a small proportion of cells express intense bursts of Tax.”

8) Fig S2 lacks a label on the x-axis. Why are the patterns so different in the top and bottom figures of each of S2 Fig and S3 Fig which presumably are replicate analyses?

9) Fig 3A is unclear – is this just a schematic without actual data? why are there 2 peaks downstream of the provirus and only 1 upstream? It is surprising that there are so manyt contact sites without CTCF sites. In Fig 3 legend what is meant by “….both in both peaks with a CTCF site…” In Fig 3C, what are each of the panels?

10) How were the genes selected in S7 Fig for analysis? Can these results be generalized?

Reviewer #2: 1) The authors show that read-through of same sense transcription in the host genome could lead to production of aberrant mRNAs. They then comment in the Discussion that read-through of same sense transcription in the host genome was rarely observed in other previous studies using ATL cells (Kataoka et al., Rosewick et al.). Could these observations be interpreted as infected clones that produce aberrant mRNAs from read-through, same-sense transcription being selected against becoming ATL cells?

2) Additionally, did the authors determine whether these aberrant mRNAs are stable? Could they be exported from the nucleus and possibly translated? A discussion regarding this aspect would be interesting since they mentioned “potential biological functions” of these transcripts in the Discussion.

3) Table 1: R and F legends are missing

4) Figure 2: In figure 2A, clone HA1 is also labelled clone 3.60. Figure 2B, C, D: 11.63 and 3.60 appear to have the same khaki color and cannot be discriminated; d2EGFP-11.50 is not visible.

5) In S6 Fig, clone 11.60 is mentioned. Should it be clone 11.63?

Reviewer #3: None

PLOS authors have the option to publish the peer review history of their article (what does this mean?). If published, this will include your full peer review and any attached files.

Reviewer #1: No

Reviewer #2: No

Reviewer #3: No
---

## [Decision Letter · Decision Letter 1]

12 Feb 2024

Dear Dr. Bangham,

We are pleased to inform you that your manuscript 'The impact of HTLV-1 expression on the 3D structure and expression of host chromatin' has been provisionally accepted for publication in PLOS Pathogens.

Best regards,

Edward William Harhaj, Ph.D.

Academic Editor

PLOS Pathogens

Susan Ross

Section Editor

PLOS Pathogens

Michael Malim

Editor-in-Chief

PLOS Pathogens

orcid.org/0000-0002-7699-2064

Reviewer Comments (if any, and for reference):

Reviewer's Responses to Questions

**Part I - Summary**

Reviewer #2: All my comments were addressed.

Reviewer #4: In this manuscript by Yaguchi et al., authors found that HTLV-1 plus strand transcription induces aberrant transcription in the flanking genome and splicing with downstream splice acceptor sites. Upon enhanced plus strand transcription, the frequency of chromatin loop formation with the host chromatin in the vicinity of the integrated provirus declined but could be recovered upon short-term application of an inhibitor of transcription elongation. This suggests that HTLV-1 proviral gene expression leads to reversible disruption of chromatin loops in the vicinity of the integrated provirus.

This study is well-written and reports both expected and unexpected novel findings. HTLV-1 integration impacts transcription of flanking host genes, which is known from other retroviruses including HTLV-1/BLV, but convincingly shown in this study using several single-cell clones from HAM/TSP patients applying innovative techniques. The study also provides examples of aberrant splicing following HTLV-1 integration. The negative impact of HTLV-1 plus strand transcription on chromatin looping is the highlight of this study and of great interest to the scientific community since it is the first report to show that not the presence of the provirus, but its transcription is responsible for disruption of chromatin looping.

**Part II – Major Issues: Key Experiments Required for Acceptance**

Reviewer #2: (No Response)

Reviewer #4: This manuscript had already been reviewed and the authors provide a detailed point-to-point response to all comments raised by the reviewers. Although I agree that it would be interesting to see whether the q4C profile of more clones than one changes upon blocking transcription elongation (Fig 5C), the data provided and the arguments raised by the authors convince me.

**Part III – Minor Issues: Editorial and Data Presentation Modifications**

Reviewer #2: (No Response)

Reviewer #4: none

PLOS authors have the option to publish the peer review history of their article (what does this mean?). If published, this will include your full peer review and any attached files.

Reviewer #2: No

Reviewer #4: No

---

## [Editor Report · Acceptance letter]

27 Feb 2024

Dear Professor Bangham,

We are delighted to inform you that your manuscript, "The impact of HTLV-1 expression on the 3D structure and expression of host chromatin," has been formally accepted for publication in PLOS Pathogens.

Best regards,

Michael Malim

Editor-in-Chief

PLOS Pathogens

orcid.org/0000-0002-7699-2064